Multiple global radiations in tadpole shrimps challenge the concept of ‘living fossils’

Mathers Thomas C. 1
Hammond Robert L. 2
Jenner Ronald A. 3
Hänfling Bernd 1
Gómez Africa 1 a.gomez@hull.ac.uk
1 School of Biological, Biomedical and Environmental Sciences, University of Hull , Hull , UK
2 Department of Biology, University of Leicester , Leicester , UK
3 Life Sciences Department, The Natural History Museum , London , UK
Crandall Keith
Electronic publication date: 2013 Apr 2
Publication date: 2013
Volume: 1
Electronic Location ID: e62
Received 2013 Feb 20; Accepted 2013 Mar 14
Copyright: © 2013 Mathers et al.
Copyright year: 2013
Copyright holder: Mathers et al.
License: This is an open access article distributed under the terms of the Creative Commons Attribution License, which permits unrestricted use, distribution, and reproduction in any medium, provided the original author and source are credited.
License URL: https://creativecommons.org/licenses/by/3.0/

Keywords: Triops, Notostraca, Lepidurus, Biogeography, Diversification, Fossil, Bayesian analysis, ‘Living fossil’, Divergence time

Funding: NERC CASE Studentship NE/G012318/1 NERC Advanced fellowship NE/B501298/1 This work was part of TCM’s Ph.D., funded by a NERC CASE Studentship NE/G012318/1. AG was supported by a NERC Advanced fellowship NE/B501298/1.The funders had no role in study design, data collection and analysis, decision to publish, or preparation of the manuscript.

==============================
‘Living fossils’, a phrase first coined by Darwin, are defined as species with limited recent diversification and high morphological stasis over long periods of evolutionary time. Morphological stasis, however, can potentially lead to diversification rates being underestimated. Notostraca, or tadpole shrimps, is an ancient, globally distributed order of branchiopod crustaceans regarded as ‘living fossils’ because their rich fossil record dates back to the early Devonian and their morphology is highly conserved. Recent phylogenetic reconstructions have shown a strong biogeographic signal, suggesting diversification due to continental breakup, and widespread cryptic speciation. However, morphological conservatism makes it difficult to place fossil taxa in a phylogenetic context. Here we reveal for the first time the timing and tempo of tadpole shrimp diversification by inferring a robust multilocus phylogeny of Branchiopoda and applying Bayesian divergence dating techniques using reliable fossil calibrations external to Notostraca. Our results suggest at least two bouts of global radiation in Notostraca, one of them recent, so questioning the validity of the ‘living fossils’ concept in groups where cryptic speciation is widespread.

Introduction

There has been much debate about the tempo and mode of the diversification of life (Eldredge & Gould, 1972; Reznick & Ricklefs, 2009; Rhodes, 1983). Recently, this debate has been informed by dating using relaxed molecular clocks and diversification analyses; techniques which have revealed disparate patterns of speciation with early bursts (Burbrink & Pyron, 2010), recent radiations (Nagalingum et al., 2011) and density dependency (Phillimore & Price, 2008) being demonstrated. One extreme and often controversial pattern of diversification is found in ‘living fossils’, a concept introduced by Charles Darwin in On the Origin of Species when dealing with the perplexing nature of the platypus and lungfish, relicts of once diverse groups (Darwin, 1859). Since Darwin’s first use, the ‘living fossil’ term has been applied to groups which appear to have diversified little and are morphologically stable over long periods of evolutionary time, with examples including cycads, tuatara, coelacanths, horseshoe crabs and Ginkgo biloba. However, morphological stasis can obscure the patterns of species diversification, and recent time-calibrated phylogenetic analyses of some ‘living fossils’ have indeed revealed that extant species are in fact only recently diverged (Kano et al., 2012; Nagalingum et al., 2011).

Notostraca, or tadpole shrimps, is an ancient, globally distributed order of branchiopod crustaceans with a rich fossil record dating back to the early Devonian (Fayers & Trewin, 2002). The order has two extant genera, Triops and Lepidurus, in the family Triopsidae, with a yet undefined number of species. The nomenclature and systematic position of some ancient extinct Notostraca lineages, is, however, problematic (Hegna & Dong, 2010). This is partly because tadpole shrimps have maintained an extremely conserved, yet complex, bauplan with extant species indistinguishable from fossils of Triops from the Triassic (Gall & Grauvogel-Stamm, 2005; Gore, 1986; Trusheim, 1938) and of Lepidurus in the Jurassic (Barnard, 1929; Haughton, 1924). This striking morphological conservatism has led them to be referred to as ‘living fossils’ (Fryer, 1988; King & Hanner, 1998; Mantovani et al., 2008).

Phylogenetic reconstructions of extant Notostraca show a strong biogeographic signal (Mathers et al., 2013; Vanschoenwinkel et al., 2012). In Triops, species complexes are largely restricted to single continents, while Lepidurus lineages show high levels of endemism (Rogers, 2001), patterns that suggest ancient radiation with diversification through continental break-up. However, the extreme morphological conservatism of this order hampers both the taxonomy of extant species and the phylogenetic placement of fossil taxa, with little known about the timing and tempo of notostracan diversification. Genetic analyses have revealed widespread cryptic species (King & Hanner, 1998; Korn et al., 2010; Korn & Hundsdoerfer, 2006; Macdonald, Sallenave & Cowley, 2011; Vanschoenwinkel et al., 2012), further illustrating the difficulty of inferring past and present diversity. To address this difficulty we infer a robust phylogeny of all known notostracan species from both extant genera and seven branchiopod outgroups. Our analysis uses all available Notostraca sequence data for seven genes, and Bayesian relaxed clock dating techniques, with multiple branchiopod fossil calibrations, to estimate divergence times.

Materials and Methods

Species delimitation

As Notostraca is known to contain cryptic species complexes (e.g. King & Hanner, 1998), and in order to follow the same criterion for species selection for the multilocus analysis, we delimited species using a generalised mixed Yule coalescent (GMYC) model (Pons et al., 2006) fitted to an ultrametric phylogeny based on all available cytochrome oxidase I (COI) sequences from GenBank. The 270 sequences were aligned with Muscle (Edgar, 2004) and phylogeny estimated with BEAST v1.7.4 (Drummond et al., 2012) under a constant population size coalescent tree model and GTR + Γ substitution model. A strict molecular clock was used with the substitution rate fixed to 1 to provide branch lengths relative to an arbitrary time scale. The MCMC chain was run for 9,000,000 iterations with the first 500,000 iterations removed as burnin. Effective sample sizes (ESS’s) of parameters (all greater than 200) and appropriate burnin were checked using Tracer v1.5 (Rambaut & Drummond, 2007). From this a maximum clade credibility tree using median heights was made. We then fitted a single threshold GMYC model to the COI tree to delimit species from populations. A total of 34 species of Notostraca were identified in this analysis (Table S1).

Multilocus phylogenetic analysis of Branchiopoda

We constructed a multilocus alignment containing representatives of all known species of Notostraca. Single representatives of each phylogenetic species identified by the GMYC analysis were selected for inclusion in our phylogenetic analysis. In addition, four Notostraca lineages (T. gadensis, T. cf. granarius Tunisia, L. bilobatus and L. cryptus), which did not have COI data available, but were represented by other genes, were also used in our multilocus phylogenetic analysis. The species status of these lineages has been confirmed in regional studies of cryptic diversity (King & Hanner, 1998; Korn et al., 2010; Korn & Hundsdoerfer, 2006; Rogers, 2001). We also included seven representatives of the other branchiopod orders so that robust fossil calibrations could be applied for the dating analysis. We included sequences for the genes 12S, 16S, 28S, cytochrome oxidase I (COI), Elongation Factor 1-alpha (EF1), RNA Polymerase II and Glycogen Synthase (see Table S2 for Accession Numbers).

Sequences were aligned using MUSCLE (Edgar, 2004) with final adjustments by eye. Introns in the nuclear protein coding genes were identified and removed based on alignment with available Notostraca mRNAs. Translation was checked in MEGA 5 (Tamura et al., 2011). Overall, sequences for 45 taxa (38 notostracan and 7 branchiopod outgroups) were concatenated for analysis with the alignment containing 5793 positions and 52% missing data (Table S2; the alignment file is available in Dryad DOI 10.5061/dryad.77bt2).

Optimum partitioning schemes and substitution models for our phylogenetic analysis and divergence time estimation were identified using PartitionFinder (Lanfear et al., 2012). PartitionFinder uses a heuristic search algorithm, starting with a fully partitioned analysis (gene and codon position where appropriate), and identifies the best fit partitioning scheme and substitution models based on Bayesian Information Criterion (BIC). Due to the restricted model choice available in RAxML (Stamatakis, 2006) we conducted separate PartitionFinder analyses for the phylogenetic analysis and divergence time estimation. For the phylogenetic analysis we restricted model choice options to GTR or GTR + Γ, whereas for divergence time estimation we allowed models to be selected from the full suite available in BEAST. We excluded models with proportion of invariant sites (+I) as rate heterogeneity is accounted for by the gamma shape parameter ( + Γ). Optimum partitioning schemes and substitution models for both analyses are given in Tables S3 and S4.

Branchiopod phylogeny was estimated using Bayesian and maximum likelihood (ML) methods with partitions and substitution models set to those identified by PartitionFinder (Table S3). Bayesian analysis was performed with MrBayes v3.2 (Ronquist et al., 2012). Model parameters between partitions were unlinked. Two independent MCMC chains were run for 10,000,000 iterations each, sampling every 5,000 iterations. The first 25% of each run was discarded as burnin with the remaining samples pooled and used to create a maximum clade credibility tree. Maximum likelihood phylogenetic analysis was performed using RAxMLHPC-PTHREADS v7.0.4 (Stamatakis, 2006). An initial ML search using GTR + Γ4 was performed onto which 100 rapid bootstraps were drawn.

Bayesian relaxed clock divergence dating

We estimated Bayesian divergence times with BEAST v1.7.4 (Drummond et al., 2012) using an uncorrelated lognormal relaxed clock (Drummond et al., 2006) and a Yule speciation prior. XML files for all BEAST runs were created using BEAUTi v1.7.4 (Drummond et al., 2012). Topology was constrained to that of the unconstrained RAxML analysis. We used the best fit partitioning scheme identified by PartitionFinder (Table S4) and estimated substitution model parameters independently for each partition. Initial runs were conducted using substitution models identified by PartitionFinder, however, this resulted in poor mixing of some GTR model parameters for partitions 1, 2 and 5, so subsequent runs were performed using a simpler HKY + Γ model for these partitions.

Table 1 Fossils used to calibrate divergence time analysis in BEAST.

Age constraints are treated as hard bounds unless otherwise stated. Node numbers indicate phylogenetic placement of fossil calibrations in Fig. 1.

Node	Fossil taxa	Geological period	Minimum age
(Mya)	Maximum age
(Mya)	Reference	
1	Oldest Bilateria eg. Kimberella	Ediacaran	-	558	Fedonkin, Simonetta & Ivantsov (2007)	
1	Rehbachiella	Upper Cambrian	-	500 (soft max)	Waloßek (1995)	
1	Undescribed anostracan	Base Ordovician	488	-	Harvey, Vélez & Butterfield (2012)	
2	Castracollis	Pragian, Early Devonian	410	-	Fayers & Trewin (2002)	
3	Ebullitiocaris elatus	Carboniferous	300	-	Womack et al. (2012)	
4	Daphnia and Ctenodaphnia sp.	Jurassic/Cretaceous	145	-	Kotov & Taylor (2011)	

Five branchiopod fossils representing the oldest known occurrences of their respective crown groups were used to calibrate the molecular clock with minimum age constraints (Table 1). Lognormal prior distributions were used to specify the level of uncertainty in the placement of these fossil calibrations as they reflect the likely scenario that the true date of divergence of a given node was some time before the earliest known fossil belonging to that clade (Ho & Phillips, 2009). Lognormal distributions have three parameters – mean, standard deviation and offset. We set the offset to correspond to the minimum age of the node as determined by the fossil record, we then specified mean and standard deviations that resulted in 95% of the distribution falling between the age of the fossil and the age of the next oldest fossil (at a lower taxonomic level) for that group. This gives a prior distribution, which assigns the majority of the probability close to the age of the oldest known fossil, but gives a long tail to account for uncertainty in the proximity of the fossil to the true date of divergence. As Bayesian divergence dating benefits from at least one maximum age constraint we conservatively constrained the root of the tree to a maximum age of 558 Mya, the age of the oldest bilaterian fossil (Fedonkin, Simonetta & Ivantsov, 2007).

We ran two independent BEAST MCMC chains for 50,000,000 iterations, sampling every 5000 iterations. Ten million iterations were removed as burnin from each run. Convergence of the two runs and the ESS of parameter estimates (all greater than 250) where assessed using Tracer v1.5 (http://tree.bio.ed.ac.uk/software/tracer/). A posterior sample of 8000 trees from one of the runs was used to construct a maximum clade credibility time tree for Notostraca and our selected outgroups. The XML file used to run the BEAST divergence time analysis can be downloaded from Dryad DOI 10.5061/dryad.77bt2.

Diversification analysis

Patterns of diversification through time within Notostraca were investigated using LASER (Rabosky, 2006) based on the BEAST time tree with outgroups pruned. Using LASER we compared constant rate and variable rate speciation models using likelihood ratio tests (Table S5).

Results and Discussion

The most complete taxon sampling to date coupled with the inclusion of multiple nuclear and mitochondrial markers allowed us to generate a robust phylogeny of 38 extant Notostraca species. ML and Bayesian inference gave highly congruent phylogenetic trees with most branches highly supported (Figs. S1 and S2). The recovered relationships between branchiopod orders are in agreement with recently published arthropod phylogenies (Regier et al., 2010; von Reumont et al., 2012), providing a solid platform for divergence dating analysis.

Our robust time-calibrated phylogeny of Branchiopoda (Fig. 1) clearly shows that Notostraca has a pattern of diversity incompatible with Darwin’s original usage of the term ‘living fossil’ as relics of once diverse groups, and importantly reveals cryptic patterns of diversification. Our analysis – using outgroup fossil calibrations – estimates an ancient divergence of Triops and Lepidurus during the Jurassic, 184 Mya (95% confidence interval 132–259 Mya), which agrees with the earliest fossils assigned to Lepidurus (Barnard, 1929; Gand, Garric & Lapeyrie, 1997), and with a sister relationship of Notostraca to the extinct order Kazacharthra of the Late Triassic/Early Jurassic (Olesen, 2009). This initial radiation of extant Notostraca was not, however, due to continental break up as the timing and pattern of diversification within the genera substantially postdates the break-up of Pangaea 160–138 Mya (Scotese, 2001). Furthermore, the current species distributions of Triops and Lepidurus are likely to have resulted from a second global radiation of the order, possibly following considerable levels of extinction. We conclude this because fossil Notostraca, attributed to Triops and Lepidurus, have been found in modern day North and South America, Europe, Africa and Antarctica, implying a global distribution by the early Jurassic (Gall & Grauvogel-Stamm, 2005; Gand, Garric & Lapeyrie, 1997; Garrouste, Nel & Gand, 2009; Gore, 1986; Haughton, 1924; Trusheim, 1938). Yet, our LASER analysis shows a significant increase in the rate of diversification of Notostraca about 73 Mya (Fig. 2), close to the time of the K-Pg mass extinction event. It is this second radiation that resulted in the current global distribution of extant Triops and Lepidurus.

Figure 1 Time calibrated phylogeny of 38 notostracan species and seven branchiopod outgroups.

Numbers at nodes correspond to the fossil calibrations given in Table 1. Nodes with black circles have ML Bootstrap support values greater than 70 and posterior probabilities greater than 95 from the RAxML and MrBayes analyses respectively. Error bars show the 95% confidence intervals of divergence times. Colour coded squares show the known geographic distribution of each species.

Figure 2 Diversification of Notostraca through time.

Arrows indicate the timing and direction of shifts in rate of diversification inferred by LASER. N is the number of species. The best fit ML model of diversification identified three distinct rates of diversification during the evolutionary history of Notostraca with an increase in speciation rate 73 Mya followed by a decrease 6 Mya.

The almost synchronous radiation of Triops and Lepidurus (Fig. 1) suggests that a common factor may have triggered diversification of the two genera. The diversification of modern birds – widely involved in dispersal in aquatic invertebrates (Green et al., 2005; van Leeuwen et al., 2012) – coincided with the initiation of the notostracan radiation (Pacheco et al., 2011), and may have facilitated the long distance dispersal and subsequent diversification of Notostraca. Indeed, the geographical distribution of extant taxa (Fig. 1) suggests several instances of intercontinental dispersal. For example, the colonisation of North America from Australia could have resulted from dispersal events during bird migration. Such long distance dispersal and colonisation events might also have been facilitated by the flexible nature of sexual systems found within Notostraca (Mathers et al., 2013). Indeed, the evolution of androdioecy – a sexual system where males and hermaphrodites coexist (Weeks, 2012; Zierold, Hanfling & Gómez, 2007; Zierold et al., 2009) – from gonochorism appears to have favoured postglacial recolonisation in the species Triops cancriformis (Zierold, Hanfling & Gómez, 2007) and in Notostraca as a whole (Mathers et al., 2013).

The concept of ‘living fossils’ has been a controversial one as it has often been interpreted to imply a lack of evolutionary change, even against evidence of molecular evolutionary change (Avise, Nelson & Sugita, 1994; Casane & Laurenti, 2013). Our divergence dating analysis has shown that tadpole shrimps can be regarded as ‘living fossils’ only on the grounds of morphological conservatism, not on the basis of limited diversification or relict status. Instead, throughout their long evolutionary history, notostracans have undergone multiple global radiations and high species turnover. Recent, time calibrated, phylogenetic analysis of other traditional ‘living fossils’ such as cycads (Nagalingum et al., 2011), nautiloids (Wray et al., 1995), horseshoe crabs (Obst et al., 2012) and monoplacophorans (Kano et al., 2012), have also revealed that extant species are more recently diverged than suggested by fossil data alone. We therefore caution against drawing conclusions about patterns of diversification based on fossil data alone in groups where widespread morphological conservatism may obscure rampant cryptic speciation. Furthermore, our results help clarify the term ‘living fossils’, putting important questions into focus. Namely, is such morphological conservatism, in the face of evolutionarily recent diversification and radiation, best accounted for by unchanging selection or by developmental genetic constraints?

Supplemental Information

Figure S1 Maximum likelihood phylogeny of Branchiopoda inferred with RAxML using a GTR + Γ substitution model for each partition.

Data was partitioned according to the best scoring scheme identified by PartitionFinder. Numbers at nodes give RAxML rapid bootstrap support values (100 replicates) with values less than 50 not shown.

Click here for additional data file.

Figure S2 Bayesian phylogeny of Notostraca based on a 7 gene supermatrix inferred with MrBayes.

Data was partitioned based on the optimum strategy identified by PartitionFinder under BIC. A GTR + Γ model of nucleotide evolution was specified for each partition. Values at nodes show posterior probabilities with values less than 50 not shown.

Click here for additional data file.

Table S1 GMYC species delimitation results.

Acession numbers within each ML cluster defined by the GMYC model. Accessions in bold were used in the multilocus phylogenetic analysis.

Click here for additional data file.

Table S2 Accession numbers of sequences included in the supermatrix.

Click here for additional data file.

Table S3 Optimum partitioning scheme and best fit models identified by PartitionFinder with model choice restricted to GTR and GTR + Γ.

Click here for additional data file.

Table S4 Optimum partitioning scheme and best fit models identified by PartitionFinder with model choice restricted to those available in BEAST.

Click here for additional data file.

Table S5 Comparison of diversification models fitted to the BEAST time tree.

Only Nostostraca and one outgroup were included in the analysis. Analysis conducted using LASER. The best scoring model is shaded.

Click here for additional data file.

This work was part of TCM’s Ph.D. We thank Chris Venditti for his help with the analyses, and Dave Lunt and Steve Moss for bioinformatics support. Domino Joyce read and provided constructive comments in a previous version of the manuscript.

Additional Information and Declarations

Competing Interests

Author Contributions

Data Deposition

There are no Competing Interests to declare.

Thomas C. Mathers conceived and designed the experiments, performed the experiments, analyzed the data, wrote the paper.

Robert L. Hammond, Ronald A. Jenner, Bernd Hänfling and Africa Gómez conceived and designed the experiments, wrote the paper.

The following information was supplied regarding the deposition of related data:

Dryad: DOI 10.5061/dryad.77bt2

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
