# Peer review of "Multiple global radiations in tadpole shrimps challenge the concept of ‘living fossils’"

_PeerJ, doi:10.7717/peerj.62_

## Round 0.1 · original submission · Minor Revisions

Two expert reviewers have now looked at your paper and both recommend revisions. Note there is significant concern with the general writing, but both feel the manuscript is appropriate for the journal. Please pay careful attention to the associated reviews.

Reviewer 1 ·

Basic reporting

No comments

Experimental design

The data used were quite extensive and the methods appropriate.

Validity of the findings

I find no issues with the methodology of the authors nor the interpretation of their results.

Additional comments

This manuscript investigates the assumption that the Notostracan ‘tadpole shrimp’ are living fossils and that they likely contain cryptic diversity. The authors use molecular phylogenetic and molecular clock techniques to build evolutionary hypotheses concerning the relationships between and diversification of these branchiopod crustacean, ultimately suggesting a recent radiation coinciding with the Cretaceous Paleogene mass extinction and the subsequent recovery & diversification of modern birds as a dispersal vector.

The article is well written although lacking reference to important and relevant works by Thorid Zierold (2006, 2007, 2009), specifically in regards to the biogeographical aspect. The size of the molecular dataset is impressive and the phylogenetic and comparative tools contemporary and interesting.

Not having any fossil taxa for any in-group nodes ultimately leads to an assumption that the rate of molecular divergence for the in group is parallel to that of the out groups. I would recommend including one of the numerous Notostracan fossil taxa, if possible, to elucidate whether this is the case. There also seems to be little investigation into how the selected node constraints affect each other. I would have liked to have seen some evidence of the authors investigating how these different assumptions affect the age of other nodes in the tree produced. I would have liked to see a brief section of the Notostracan fossil record, some major works and fossil forms related to extant taxa perhaps (for instance Hegna & Dong, Acta Geological Sinica, 2010, 84:886-894). This would provide a better context for the use of information from the fossil record.

I think this manuscript is very interesting and of great value to understanding the evolutionary biology of branchiopod crustacea, as well as challenging the biological concept of ‘living fossils’. I would suggest publication with minor revisions

Specific comments:
1) The conclusion on lines 151-153, of recent radiation of cryptic species, does not detract from the ‘stasis’ seen in the parent lineage; the age of these ancestral lineages suggests some degree of stasis, lending credence to the idea of these crustaceans being ‘living fossils,’ in my opinion.
2) Lines 27-28, example taxa, should use Linnaean binomial?
3) Line 50, Notostraca species or Notostracan species?

Reviewer 2 ·

Basic reporting

Overall the manuscript is reasonably well written, not stellar, but acceptable. It does need to be re-read for grammatical problems correcting sentences that are opaque. E.g., line #23, remove "that". Sentence #lines 28-30 not clear. Line 54, beginning section heading with abbreviation is awkward. Line 76 and 84, vernacular “branchiopod” is lower case; if “Branchiopoda” then upper case.
Line 67, Table S1, not provided, cannot be evaluated. Similarly, line 86, line 98, Table S2-S4, cannot be evaluated. Line 111, BEAUTi v1.7.4 needs citation.

There is sufficient introduction and background with the exception that there is no mention of how many species/genera there are today. Relevant prior literature is referenced. Two figures and one table are provided. Supplementary tables necessary to assess results were not available. The ms. does represent an appropriate ‘unit of publication’.

Experimental design

Based on the terminal taxa in Figure 1, taxon sampling appears adequate, fossil taxon sampling is comprehensive, outgroup selection is good. I am unfamiliar with GMYC model and can not comment on the analysis methods. I defer here to other better qualified reviewers than me, ones who can better evaluate analysis methodology and fossil placement. Based on my poor understanding of the methodology, I presume once cryptic species complexes are discerned a subset of sequences are aligned for the multiple genes. It seems this could be stated in a more straight forward manner.

Validity of the findings

I do not have the expertise to evaluate the robustness and appropriateness of the methods used -- which precludes my evaluation of the authors speculations. In Figure 2 is “N” the number of species, genera, or other taxon level? There is no mention in the ms. of the diversity of notostracans today. All I see are the terminal nodes on the tree.

Additional comments

No comments.

---

## Round 0.2 · accepted · Accept

I applaud you for submitting your alignment matrix to Dryad. I would ask that you go one step further and submit your resulting tree(s) to TreeBase. This will also other researchers the capacity to access your phylogenies for other purposes. Thank you for responding efficiently to the reviewers' comments.